# Which Language Evolves Between Heterogeneous Agents? - Communicating Movement Instructions With Widely Different Time Scopes

**Marie Ossenkopf**
Distributed Systems
University of Kassel
Kassel, Germany
marieossenkopf@gmail.com

**Kevin S. Luck**
Finnish Center for AI
Aalto, Finland
ksluckml@gmail.com

**Kory W. Mathewson**
DeepMind
korymath@gmail.com

## Abstract

This paper studies the evolving communication between two agents, a listener and speaker, in a plan execution task in which the speaker needs to communicate the plan to the acting agent, while operating on different time scales. We analyse the topographic similarity of the resulting language learned by the proposed imagination-based learning process. As the speaker agent perceives the movement space strictly in absolute coordinates and the actor can only choose relative actions in the movement space, we can show that the structure of their emergent communication is not predestined. Both relative and absolute encodings of desired movements can develop by chance in this setting, but we can alter the chance by using a population of learners. We conclude that our imagination-based learning strategy successfully breaks the strict hierarchy between planner and executioner.

## 1 Introduction

The question how language shapes perception has been discussed in the field of emergent communication for quite some time now. Maybe because the idea that someone cannot perceive the color blue or count to 7 because they are lacking the words is mind dazzling (Frank et al., 2008). In contrast, the opposite constellation - how perception influences the resulting language - has been underexplored. The correlation, however, stops being obvious once the communication partners have different perceptions of the world and need to establish a shared language. Regarding reinforcement learning tasks where agents need to communicate about positions and movements in space, the predator-prey task has been a prominent example. In most realizations so far, the agents shared the same perception, thus the resulting protocols were not ambiguous (Sukhbaatar et al., 2016; Werner & Dyer, 1993; Wang et al., 2020).

In this study we investigate the outcome of the learning process of two heterogeneous agents regarding their perception of time, space and actions, and found that the emergent language is not pre-determined and accommodates one of the two agents more than the other by chance. Inspired by the predator-prey task, we work with two originally hierarchical agents, one able to overview the whole situation and create long-term movement plans and an acting agent able to follow instructions. However, the acting agent also needs to swiftly react to unforeseen changes at individual time steps, requiring both agents to learn together as equals rather than master and subordinate.

In this work, we make the following contributions:

- We present an imagination-based learning setup for two communicating agents with very different time scopes in a spacial plan execution task.

- We analyse the resulting encodings both qualitatively and topographically.

- We show that the setup can result in encodings of positions that are sometimes more relative and sometimes closer to absolute in the emergent communication of movement instruction.

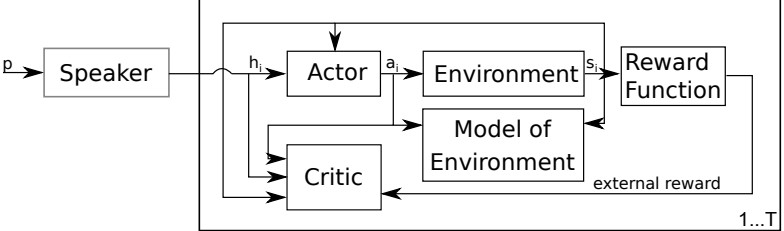

Figure 2: Forward pass in the communication set-up using an actor-critic for the trajectory execution. The critic tries to smoothly approximate the reward structure of the environment for the inputs and resulting actions of the actor.

## 2 METHOD

The agents are acting in different time scopes, with the speaker agent formulating plans each episode and the listener agent moving in the environment each time-step. These vastly different time scopes of speaker and actor require a special training setup, based on shared imagined episodes. We chose this setup to guarantee the heterogenity of the task perception between speaker and actor. For two agents acting roughly in the same environment and only with different perceptions, it might be to easy to find a mapping which would make the communication task trivial again.

Fig. 2 shows how during an episode there is one forward pass of a path through the speaker, while the actor operates in a loop to process every of the messages the speaker produces sequentially over the time of an episode. During this, the critic collects each state, message, action and reward in a replay buffer. With this, the actor is trained with off-policy reinforcement learning, here deep deterministic policy gradient (DDPG), after every episode, in conjunction with training a predictive model of the environment. We use the learned environment model to enable imagination-based training of the agents, inspired by earlier work in control such as Piergiovanni et al. (2019). First the speaker creates a set of messages with its current policy for a memorized path. The speaker then asks the actor to imagine acting out this set of messages in its model of the environment. With the imagined positions of the actor, the speaker calculates gradients from the difference of the positions the speaker intended the actor reach and the positions the actor imagined to reach (see Fig.9). This lets the speaker adapt its messages to the current understanding of the actor and creates enough meaningful feedback for the speaker to train its policy. For every training episode acted out in the environment, we let the actor do 200 updates with DDPG and the speaker dream and update 300 times with the actor and its model of the environment. The algorithm design and the underlying design decisions are explained in detail in the appendix.

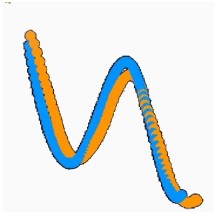

Figure 1: Example input path in blue and the positions reached by a trained actor in orange.

## 3 EXPERIMENTAL SETUP

The environment has a size of $100 \times 100$. All sides of the field are surrounded by walls that simply block any movement in that direction. The actor observes its absolute position and can choose a movement action in x and y direction with a maximum euclidean length of 1. Once per episode, the speaker receives a path $p$ in form of a cubic spline represented by its knots. From this it forms 100 continuous two dimensional messages $h_t$ and sends them sequentially to the acting agent. The acting agent receives one message per time step and, together with its current state $\mathbf{s}_t$ executes an action using its policy $\pi(\mathbf{s}_t, \mathbf{h}_t)$. The reward function is defined by the euclidean distance to the next interpolated way point on the path $p$. As we see, the action of the speaker spans the horizon of a full episode, while the actor experiences the full 100 time steps.

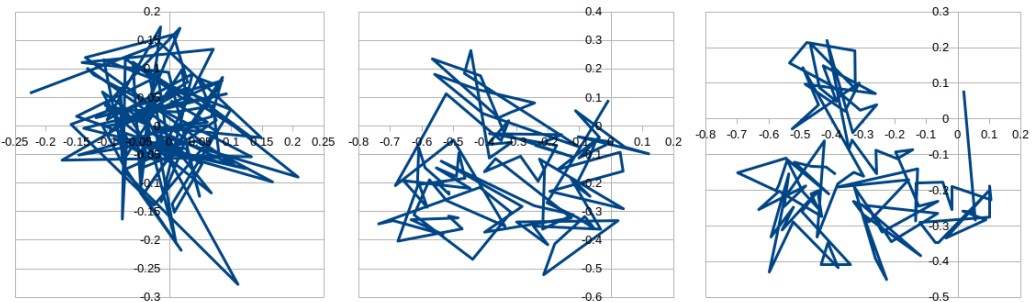

Figure 3: Messages describing the same reference trajectory produced by a speaker after 0, 300 and 500 training episodes with a single actor.

## 4 RESULTS

Initially, we analyzed the resulting languages qualitatively to get an idea of how the messages encode the positions or movements, the actor needs to reach or make. The first promising finding is that training with imagination-based internal gradients in comparison to training the speaker with the classical REINFORCE (Williams, 1992) approach allows the speaker to still receive a valuable learning signal, even if the actor does not improve on the task anymore. Fig. 3 shows the sequence of messages produced by the speaker for the same trajectory at different points in the training. In this example, the actor's performance stopped improving somewhere between training episodes 100 and 200.

What we see here, is that the messages to describe the trajectory are becoming more and more ordered. One can imagine, that the three visible clusters belong to the different knots of the spline, which is the path. This ordering could be resembling an absolute encoding. Positions that are close to each other are described by messages, which are close to each other, opposed to describing actions of the actor. If this ordering has in fact to do with the absolute positions of path points, it can be measured by the topological similarity between the trajectory space and the message space, as has been proposed by Brighton & Kirby (2006). In those terms, a high degree of ordering means that the distances between two points in the trajectory space correlate with the distance between the corresponding messages. As the task at hand resembles more a reconstruction game than a referential game, we expected to find a high topological similarity in line with the research of Guo et al. (2020). We measure the topological similarity as the Pearson correlation between the distance of two random points in different trajectories and the distance between the messages used by the speaker to direct the actor in the respective time step.

Although we found examples like 3, in most examples, the topological similarity in Fig. 5 only rises over the first 150 learning episodes and decreases again afterwards. This is a behavior that has been observed too by Ren et al. (2019) when searching for compositionality in emergent protocols for reconstruction games. After a certain period of uninterrupted interaction, two agents may start to develop individual protocols that diverge from logical structure. We see, however, that the actor's performance in reaching the desired trajectory points still increases steadily, showing effective communication learning.

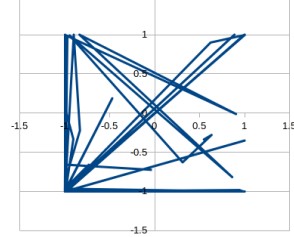

We conclude that the spacial structuring of the messages according to the trajectory, as we can measure it with topological similarity, is not indicative of successful communication of the trajectory. The speaker finds other ways to direct the actor's movements than encoding absolute positions in space for every time step. Note that

Figure 4: Description of a reference trajectory produced by the speaker at the end of the training process from Fig. 5.

topological similarity would not be able to detect relative position statements like "up" or "left" because they do not show any correlation with the positions in trajectory space. In Fig. 4, we can see the series of messages belonging to the end of training in Fig. 5. The speaker mainly just uses

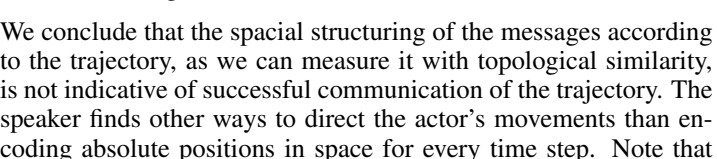

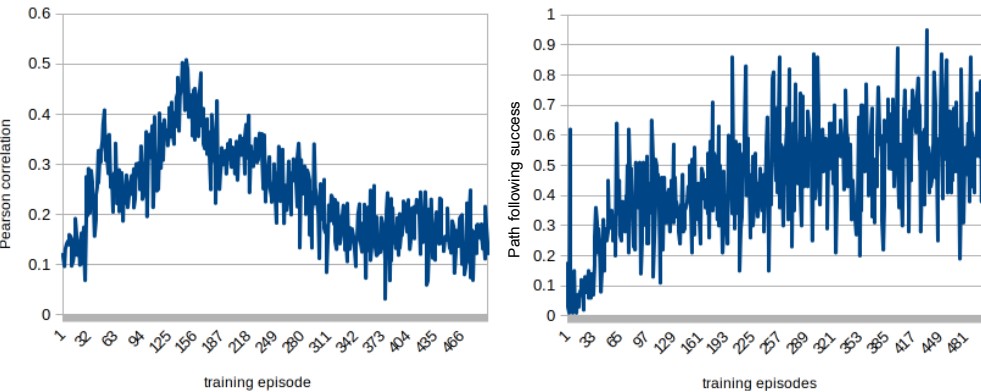

Figure 5: The topological similarity between trajectory and message space over the learning process of a single unreset pair of speaker and actor and the path following performance of the actor during the same learning process.

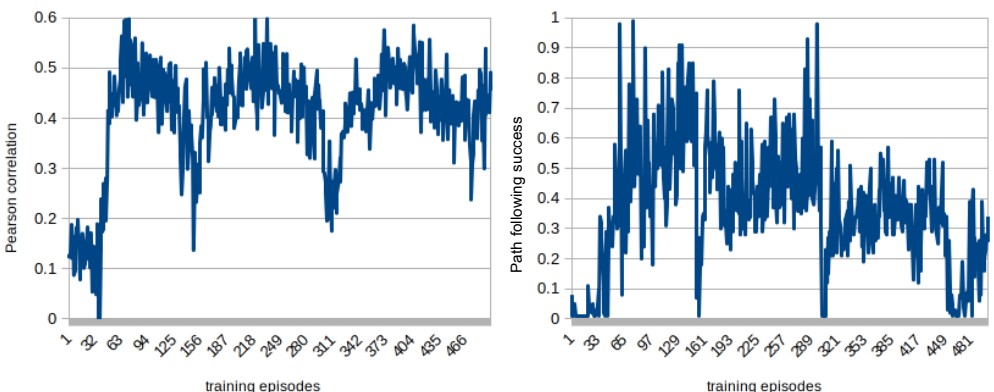

Figure 6: The topological similarity between trajectory and message space over the learning process for a population of actors where a newly initiated actor is added every 150 training episodes and path following performance of the actors during the same learning process.

four clusters of messages ([1,1],[1,-1],[-1,-1][-1,1]), probably comparable to the distinct words "up," "down," "left" and "right," which we would use to direct someone through a 2D world. Considering the actor's high performance in matching the trajectory we see in Fig. 5, this seems to be an efficient approach, although we cannot clearly deduct the meaning of the protocol. Note that we also encounter a lot of examples that do not look so distinctly like an absolute encoding like 3 or like a relative encoding like 4. They might be mixtures, that we cannot distinguish with topographic similarity yet. This gives us to the open question, how this kind of inherent structure could be measured other than by topographic similarity, as well as if there are possibility to tweak the tasks of heterogeneous agents to favor certain encodings.

Ren et al. (2019) proposed that adding newly initiated agents to a population can introduce a bias towards a more structured language. If we employ an approach with a population where we introduce a new randomly initiated agent to the population of actors every 150 training episodes, we see that we can keep the topological similarity high after the first peak. In Fig. 6, we can, however, also see that the performance on following the trajectory decreases with each actor introduced and does not reach the previous actor's level.

## 5 DISCUSSION

An absolute encoding is easier for the speaker, because it already receives its input in absolute coordinates. A relative encoding on the other hand is easier for the actor, because it needs to output its actions as relative movements. Both are, however, able to accomodate to the other with additional calculation cost. As we observed encodings in both directions, the power of the agents seems to be somewhat equally distributed as not one of them overpowers the other with its preference and rather the initialization leads to one or the other encoding. Tweaking the power dynamic between communicating agents could have interesing effects on the resulting languages.

## 6 CONCLUSION AND OUTLOOK

We presented a draft for an imagination-based algorithm design to train two agents operating on very different time scales and still evolving a shared protocol together. With this setup we observe not one predetermined structured of the evolved encoding, but two extremes which accustom either one of the communicating agents more. If both agents have equal power on the emergent communication, mixtures of both encodings have a chance to show up. One main takeaway is that a topographic structure only shows if the language follows the structure of the absolute perception space, but this is not necessary for communication success or structure in the language. We might need other means of measure to capture these kinds of relations. Also although using a population of listeners can improve the structuring of a language, that does not necessarily improve the task performance.

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

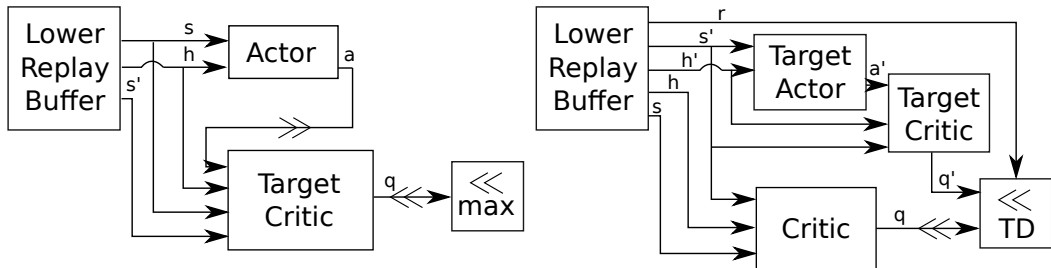

Figure 7: DDPG training scheme for actor and critic. The parallel arrows represent the backwards propagation of gradients. The actor is trained by maximizing the q-value. The critic is trained by minimizing the temporal difference between the observed reward and the estimated q-values.

# A  APPENDIX

Training two communicating agents that operate on completely different time scales and are supposed to learn together as equals is a difficult task for the algorithms design. We will comprehensively explain the reasoning behind the design decisions in the following.

## A.1  STATE-LESS ACTOR

Ossenkopf et al. (2019) have shown how difficult it is for an recurrent neural network (RNN) to follow even short execution plans. The purpose of RNNs is to smooth noise or accumulate knowledge. They are rather unsuitable for remembering a long execution plan and working through it step by step. This is like walking blindly and counting steps along a defined route. Therefore, we chose another approach than RNNs to let agents learn execution plans. A state-less actor just learns how to optimize the current action, given the current state and message. This also allows for training more complex actors with offline learning in an actor-critic network like DDPG when training examples are costly to obtain. Hence, the longer the execution plan or the more complex the actor, the more it makes sense to use a state-less actor for learning the execution. With a state-less actor, we need another entity to define goals for the actor to follow.

The speaker can be understood as breaking down a path $p$ into messages $h_j$ for the executive policy. This resembles breaking down "Move up three times, then left, then right" into ["Move up," "Move up," "Move up," "Move left," "Move right"]. The executive policy then tries to infer an action given one of these instructions at a time without remembering the rest of the message. The representation of messages as desired absolute positions offers one solution to the presented problem but the optimal solution might differ for different actors. If we train the presented model from end to end, the values of $h$ are free to converge to something completely different and more effective when made subject to optimization. How these messages turn out is one of the interesting questions in the field of emergent communication and will be discussed in detail in the next section. The resulting degrees of freedom also impose new difficulties to the algorithm, which we address comprehensively in the appendix.

Fig. 7 shows how the actor and the critic are trained using examples stored in a replay buffer. DDPG works offline, i.e., it works with the assumption that the received rewards can be approximated in a Q-value function from the observable state-action pairs. This assumption holds when the input h represents fixed messages for the whole training time. For this new set-up, h will keep changing its meaning, because the speaker network will still learn how to transform the received message into a meaningful representation. We thought, this would impose problems on the actor and critic, because the meaning of messages changes. But so far, we haven't seen anything that supports this apprehension. It might, however, be something to keep in mind as a source of learning instabilities.

## A.2 Inverted Reward Assignment

The reward structure is essential in shaping the cooperative exploration strategy in reinforcement learning. Both reward assignment as equal communication partners and as hierarchical command and execution agents are viable/reasonable approaches in trying to learn to communicate plans. In classical hierarchical reinforcement learning (RL) approaches, the upper hierarchical layer takes care of optimizing the task reward, while the lower layer only tries to follow the goals defined by the upper layer. This is a valid approach and has been shown to be successful in many tasks.

In our case, the agents are designed as equals with different tasks, but what does that mean in particular? The speaker's task is to make sure that the actor follows the plan the speaker produced. This leaves the actor both with less knowledge about the task than the speaker and the speaker with less control over the execution than the actor. Therefore, a strict hierarchical reward assignment might be off. We want to try an inverted approach instead to give the actor more freedom in optimizing the task while the speaker only focuses on keeping the actor on track. Thereby, we try to produce a more flexible combination of cooperative and hierarchical reward patterns to create synergies between different time scopes as well as actuation optimization.

## A.3 Dreaming Feedback Loop Through Actor And Speaker

One of the most important questions in reinforcement learning is: where do the gradients come from? The reward received in a single example does not allow for an estimation of the reward landscape to learn the right direction for adapting the policy. So many algorithms use bootstrapping by sampling from the current policy to estimate the gradient for a gradient ascend approach. This is especially hard in a multi-agent setting, where the changing behavior of the other agents leads to a non-stationary reaction of the environment to an agent's own actions. Some authors in emergent communication solve this problem by back-propagating gradients from the listener to the speaker through the message channel. Others see this as a violation of the multi-agent paradigm and a hindrance to extending emergent communication settings to heterogeneous agents. Both points are valid. Regarding our set-up, the speaker and the actor can be seen as two agents, as they follow different tasks and learning schemes, however, we do not consider back-propagating gradients from the actor to the speaker through the messages a break with the multi-agent paradigm, as they are just a form of complex feedback.

Using a bootstrapping approach like REINFORCE (Williams, 1992) to train the speaker is destined to fail due to the huge difference in time scopes between speaker and actor. With trajectories consisting of 100 steps, the speaker receives only one online example for every 100 examples of the actor. At the same time, the action space of the speaker has a 100 times more dimensions than the actor, which leaves the speaker more than a 10000 times worse off than the actor regarding necessary samples.

Another possibility would be to treat speaker and actor as sequential models and train them in one big actor-critic set-up. But this would require the critic to understand the reward structure for every action of the actor given the whole trajectory information as input, and we have concluded in previous experiments, that this is a hard mapping to learn.

This leaves us with the smaller actor-critic loop including only the actor that learns to follow single messages as depicted in Figure 2. We can follow the arrow representing the flow of $h$ in this scheme to get an idea from where the speaker can receive gradients. The first possibility that jumps to the eye is receiving gradients from the actor. But the speaker cannot receive gradients from the actor-critic loop because the actor does not receive its reward directly from the environment but it receives gradients from the critic (see Fig. 7). These gradients from the critic are conditioned on the messages, so there can be no information gain by back-propagating them from the actor to the speaker. The meaning would be something like: "I would have received a higher reward for this action if you had given me another message," which, however, does not tell the speaker that this other message would improve trajectory-following. Instead, it says the random action taken by the actor would have been described by another message.

Secondly, the critic conditions on the messages, so it could also back-propagate gradients to the actor. But this can only be done in online learning, because the graph of the speaker that created the messages from a certain trajectory needs to be existent and only the temporal difference regarding

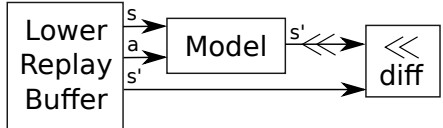

Figure 8: Training of the environment transition model with the same replay buffer as used for training actor and critic.

the reward from this trajectory yields information about how these messages affected the prediction. The critic's feedback to the speaker can then be understood as "If you had given me these messages, I would have better understood what reward to expect." This gives us meaningful gradients for the speaker to improve the set messages, but, as it is restricted to online learning, we lack the possibility of batch normalization and have to work with very few samples. So we still need a better way to train the speaker.

This is the point where the inverted reward assignment comes in handy. As we defined the task of the speaker as only making sure that the actor stays on the trajectory, and we also already learned to decode the trajectory from the message of the speaker, we can use the distance to the trajectory to produce gradients. The distance to the trajectory depends on the state reached by the agent, so, to back-propagate gradients from the distance to the speaker, we need the following calculation chain.

$$speaker \xrightarrow{message} actor \xrightarrow{action} (?) \xrightarrow{state} distance \tag{1}$$

If we compare this to Figure 2, the place of the question mark is taken by the environment. But we cannot back-propagate through the environment, as we cannot expect to know its inner workings. This makes another module necessary, because the actor alone does not know the results of its actions. We need to model the transition function of the environment. This can be done with the same replay buffer used to train the actor and the critic, because the model only needs to learn a prediction task from state and action to the successor state (see Fig. 8).

Now, this model allows us to roll out state-action trajectories offline. This means we can estimate the behavior resulting from the interaction between speaker and actor without the need for an online sample. When we calculate the gradients that minimize the distance between the states the actor reaches and the desired trajectory, we can feed them backwards through the model and the actor to the speaker. Notice that the model and actor work in a loop and, therefore, the gradients could also be fed through the model and the actor again and again for every step of the roll-out. As the actor's reaction to the first message influences every state that follows, the gradients of all states would accumulate in $h_1$, making it hardly meaningful. One typical approach to train an RNN on long trajectories is to only apply gradients to a small part of the trajectory, to avoid delusion or explosion.

We adopt this approach and will only feed the gradients of a small amount of consecutive states through model and actor. This has the effect that each message is mainly evaluated on the result it produces on the next handful of time steps, which is reasonable because the long-term planning has to be handled by the speaker's model. The speaker only has the task of breaking down the given plan into digestible bits for the actor and optimizing its messages in a slightly longer horizon than the actor. You can see the resulting training scheme if the speaker works with a horizon of a single time step in Fig. 9, where the gradients are passed through model and actor once but not through the loop that led to the current state in the roll-out. $n$ is sampled uniformly through the training process to create equal amounts of training data for each message. The gradients the speaker receives from the actor in this loop can be interpreted as: "If you had given me that message, I would have stayed closer to the desired trajectory." Note that we do not train the actor with these gradients. Rather, we let the speaker adapt to the current message understanding of the actor.

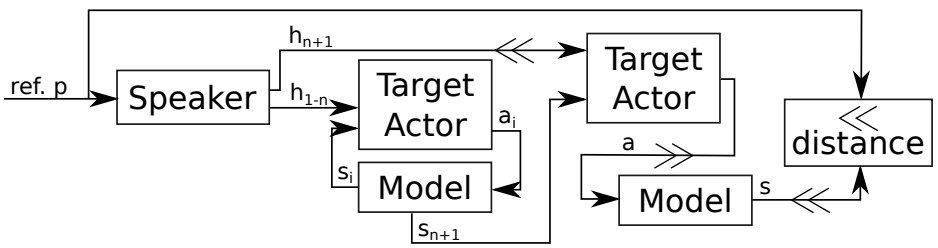

Figure 9: Training of the speaker through the custom feedback loop. Actor and model are used to roll out a state trajectory until step n. The action and state in time step n+1 allow back-propagation of gradients to minimize the distance to the reference path to the speaker.

