# OpenReview forum: "Which Language Evolves Between Heterogeneous Agents? - Communicating Movement Instructions With Widely Different Time Scopes"
_ICLR.cc/2022/Workshop/EmeCom — EmeCom Workshop at ICLR 2022_

### Official Review · Reviewer_4G3t · 2022-03-23
**Review of "Which Language Evolves Between Heterogeneous Agents?"**

**Rating:** Weak accept
**Confidence:** 4

**Review:**

## Summary
This paper presents an interesting approach for incorporating emergent communication into a hierarchical reinforcement learning problem. The system involves two agents: a speaker and an actor. The speaker can be thought of as a *planner* and the actor as an *executor*. The speaker is given a "problem" for which it must construct a sequence of messages that the actor will receive during execution. The actor is stateless, meaning that it only sees its current observation and the message prepared for it by the speaker.

The environment that the agents are trained in consists of a continuous 2D plane in which the actor must move around and follow a path defined by a cubic spline for 100 time steps. The speaker is given the knots of the path as input, and must output 100 messages - one for each time step that the agent will execute. Reward is defined by the distance that the actor is from the path after a given action.

The actor is trained by reinforcement learning (actor-critic DDPG) on the rewards directly. Loosely, the speaker is trained based on the actors ability to "follow a desired trajectory". However, this is not straight-forward as the distance between the actor's actual trajectory and the desired trajectory is not differentiable with respect to the speaker's parameters.

This is where the "imagination-based learning" comes into play. The authors train an action-conditioned state regressor that can be used as a differentiable estimator of the position that the actor lands in as a result of their action. Thus, the distance between this point and the target trajectory can provide gradients for the speaker's parameters.

The authors then present the results for training these agents, and analyse the emergent protocol using both qualitative discussion and quantitative measurement of topological similarity.

## Positives
- The paper raises the interesting question of "how perception influences the resulting language" and applies it to a situation of agents with different input and action spaces learning to communicate.
- The proposal for training the speaker using the imagined trajectory of the actor is interesting and well-motivated (in the appendix).
- The paper provokes pressing questions about interpreting the topological similarity between a message space and and the state space of a listening agent, even in a simple environment with a limited listener (in this case stateless).
- The paper demonstrates the evolution of the message protocol throughout the training, and shows that the protocol can vary.

## Criticisms
My main criticisms for the paper can be broken into two categories: (1) lack of context or explanation, (2) overstating results and using unnecessary jargon. But first, some miscellaneous points:

- The predator-prey task is mentioned twice, but its relevance to the task in the paper is never really explained.
- It is not clearly explicated how the "imagination-based method" is helping address "how perception influences the resulting language". The only part the imagination seems to play is in providing the training signal for the speaker - would this not be independent of whether or not the speaker and actor had different perception architectures?

### Lack of Context or Explanation
The paper does not properly introduce relevant work or provide the reader with useful background concepts or terminology. For instance, briefly defining what "imagination-based learning" and "dreaming" are, and more carefully explaining how they are used in the approach would be immensely helpful to the reader.

Another example is the confusing manner in which the, in the main body of the paper (section 2), the authors write:

> The speaker asks the actor to imagine acting out a set of messages in its model of the environment and then calculates gradients from the difference of the positions the speaker intended the actor reach and the positions the actor actually reached

This phrasing is very confusing as it does not clearly state that the speaker is learning from *imagined* "actual" positions that the actor reaches. It was not until going through a detailed reading of the appendix that I was able to properly understand the purpose of the imagination-based learning and its role in training the speaker. Given how crucial this is to the work, it should be much more clearly presented in the main body of the paper.

### Overstating Results
The paper overstates its results in a vague and jargon-laden manner, for instance, claiming that:

> (the) strategy successfully breaks the strict hierarchy between planner and executioner
- This claim is over stating the fact that the emergent protocol does not seem to favour easy interpretation by the actor nor easy production by the speaker.
- Firstly, I do not see why either of these should be favoured by default, but it seems to me that the authors are presuming that the speaker being on top of the hierarchy should imply that their job is "easier". I see no reason why this should be the case, and it seems like anthropomorphising.
- Secondly, I do not see how the speaker's job being harder "breaks the hierarchy" in any sense - the speaker is still dictating action to the actor.

> both relative and absolute encodings of desired movements develop
- This is claimed in the abstract, but it is not demonstrated, only speculated, that relative movements are signalled by the speaker. This is supported by the authors own words where they state "we cannot clearly deduct the meaning of the protocol" after speculating about relative movement messages.
- It is unclear to me why the authors could not back this speculation up by simply looking at the topological similarity between the action space of the actor and the message space. In other words, the Pearson correlation between the distance between two random relative direction vectors at points on the trajectory, and the distance between the messages used by the speaker to direct the actor in the respective time step.

> we can show that the structure of their emergent communication is not predestined

and later

> we observe not one predetermined structured of the evolved encoding, but two extremes which accustom either one of the communicating agents more.
- Firstly, it is not convincingly demonstrated in the paper that "both extremes" emerge (see previous criticism). Instead, a more accurate version of these claims might be "we observe that throughout training (and across runs) different 'kinds' of communication protocols emerge, as characterised by different topological similarity to the absolute position space".
- Further, to make this claim more strongly, it would be worth looking at the different reward/loss values associated with different 'kinds' of protocols. If a different kind of protocol is simply the result of underperforming it may be considered less interesting.

---

### Official Review · Reviewer_pUyx · 2022-03-23
**Heterogeneous communication review**

**Rating:** Weak accept
**Confidence:** 4

**Review:**

Summary

This paper studies the evolving communication when the two agents are communicating at different time scales. An imagination-based learning setup is proposed correspondingly.
Experiments show that the relative and absolute encodings can develop by chance and the chance can be altered by using a population of learners.

Main review

- This work proposes a new angle to study emerged communication between two heterogeneous agents and how different perceptions bias a shared language. The imagination-based learning framework is introduced to help the training under this novel setup. Using a population of learners also adds interesting aspects to the experiments.
- The experiments design can be interesting for two heterogeneous agents but the current results do not show the advantage of communicating at two time scales. A baseline like two agents communicate step by step may be added for performance comparison.
- The current analysis focuses only on absolute/relative encoding. This entangles the influence of time, space, and action on evolved communication. 1) Space: the agents are both perceiving states in absolute position, a more reasonable setting could be that the speaker perceives from an absolute view while the actor observes the states from its own coordinates. Otherwise, the encoding mixes the influence of action and space. 2) Time: similarly, when the agents communicate at two time scales, more analysis can be added to study how these two agents compromise between these temporal scales. For example, will the 100 messages mean 100 actions at each time step, or do they only includes actions at several time points?
- Since the actor’s action is also continuous, could the correlation between the distance of two actions and the distance of two messages be computed for the degree of relative encoding?
- Fig 6 shows an interesting result that a more structured language cannot successfully pass down to the next population. Comparing the step-by-step communication with the imagination learning setup may help study how the learning framework affects the results.

---

### Decision · Program_Chairs · 2022-03-25

**Decision:**

Accept

**Comment:**

Both reviewers are confident that this work should be accepted and we look forward to discussions on communication and time scopes.